# Prevalence and mechanisms of anterior cruciate ligament tears in military personnel: A cross-sectional study in Iran

Seyyed-Mohsen Hosseininejad [1,2], Mohammad Kazem Emami Meybodi[1], Mehdi Raei[1], Alireza Rahimnia[1,3]*

1 Baqiyatallah University of Medical Sciences, Tehran, Iran, 2 Orthopedic Department, School of Medicine, Golestan University of Medical Sciences, Gorgan, Iran, 3 Orthopedic Department, Taleghani Hospital, Shahid Beheshti University of Medical Sciences, Tehran, Iran

* alireza_rahimnia@yahoo.com

**Data Availability Statement:** All relevant data are within the manuscript.

**Funding:** The authors received no specific funding for this work.

## Abstract

### Background

Anterior cruciate ligament (ACL) tear is common in military setting; such an injury increase institutional costs and workforce strain, however, few studies have investigated the mechanism and associated factors of ACL tear specifically in a military setting. The aim of this study was to investigate the prevalence and mechanisms of ACL tears in military personnel at a military referral hospital in Iran.

### Material and methods

This cross-sectional study examined 402 military personnel who presented with knee complaints at a single referral Iranian military hospital. The ACL injury prevalence and mechanisms were assessed by physical examination, medical records, and magnetic resonance imaging (MRI) of the knee. Data were collected by an orthopedic resident.

### Results

Of the total 402 patients, 285 were diagnosed with ACL tears; the prevalence was 70.9%. The most common mechanism leading to ACL tear was noncontact events. The knee changing direction-knee pivoting (54%) was the most frequent lower limb status, followed by a fall with the knee in valgus position (20.7%). The most commonly associated activity was military training (63.9%) and sports activities (32.6%). The incidence of ACL injuries was higher in soldiers compared with officers during military training, but higher in officers during sports exercises (P = 0.002). Common associated injuries involved the knee meniscus and cartilage.

### Conclusion

The findings support those of previous studies, that in military personnel, the most common knee injury is damage to the ACL, most frequently through noncontact events, specifically

**Competing interests:** The authors have declared that no competing interests exist.

knee pivoting, during military activities rather than sports and among soldiers. These findings help develop ACL injury prevention programs.

## Introduction

Knee injuries are common in young athletes. Knee injuries, particularly injuries to the anterior cruciate ligament (ACL), meniscus, and articular cartilage, account for the majority of these injuries and prevent high-level athletes from maintaining peak performance. The military population, with their repetitive strenuous physical activities during their careers, should also be considered like elite athletes in terms of required fitness, in addition to performing combat exercises that define military practice [1–3]. There are reports indicating an incidence of 2.1–31 ACL injuries per 1000 individuals, which is nearly 10–14 times higher than in the general population [4–9]; Each year, 2500–3000 cases of ligament reconstruction surgery, especially for ACL, are performed in the armed forces. These enormous numbers result not only in a large number of patients, but also in a loss of military forces and skills in warfare. ACL injury can lead to posttraumatic knee osteoarthritis at a rather younger age [10–20], it could be a physical and economic burden to society [13, 20–31]. This is particularly noteworthy in military service, as ACL injury can lead to termination of perfect military performance and early retirement. Loss of military strength and warfare is inevitable thereafter [11, 32, 33]. Therefore, it is strategically important not only to provide functional care but also to prevent injuries [13, 34–36]. The ACL can be injured in different ways. Commonly, mechanisms are categorized as noncontact and contact trauma. Given that 88% of ACL injuries followed noncontact knee injuries. It is critical to recognize common mechanisms of injury and take appropriate action to reduce the risk of ACL injury [11, 12].

Previous literature from the sports field has examined the injury mechanisms and risk factors associated with ACL injuries and found that the use of prevention plans implemented during the analysis was effective in preventing ACL injuries [13–17, 21].

However, military personnel are described by extremely rigorous physical activities and demanding missions, and there is an apparent lack of reports specifically focusing on ACL injuries in the military environment, particularly in Iran. Determining the patterns of knee disease, affected groups, and activity-related risk factors will help practitioners and researchers modify the environment to identify and mitigate the risk associated with knee activity [22, 23]. Thus, the aim of this cross-sectional study was to evaluate the prevalence and mechanisms of the ACL tears in military personnel presented at a military referral hospital in Iran in order to provide data and clues for developing future injury prevention plans.

## Material and methods

### Ethics statement

The study has been approved by the local institutional review committee of Baqiyatallah University of Medical Sciences, Faculty of Medicine; the need for informed consent was waived by the ethics committee because the data documented for each patient were all applied in routine history and physical examination and could be found in medical files of patients. All methods were conducted in accordance with the ethical standards of the declaration of Helsinki.

(**Ethics Registration ID**- Baqiyatallah University of Medical Sciences: **IR.BMSU.BAQ. REC.1401.091**).

## Study design and population

This descriptive-analytic cross-sectional study examined patients who presented to a single referral military hospital from January 2022 to March 2023 with knee either complaints of pain, giving way, and locking. Exclusion criteria were: previous limb surgery; associated fracture of the affected limb; non-acute- cases with a tear that is older than six weeks at the time of their hospital presentation-; inability to determine the timing of injury based on patients history and magnetic resonance imaging (MRI) data; and inadequate medical records. After census sampling, within 402 patients presented with knee related problems, of which, a total of 285 patients aged 18–51 years in whom clinical and imaging examination confirmed the diagnosis of ACL tear were included in the study.

## Investigations

Data collection included the following: (1) demographic information-age, sex, body mass index (BMI), marital status, military rank, and injured and dominant side of the knee; (2) mechanism of injury during the event; (3) associated chondral and meniscal lesion as detected by MRI; and (4) type of activity in which the injury occurred-military training including basic and speciallized combat training- and time from onset of activity to injury. All data were kept confidential during and after collection. According to patient's recollection and clinical examination matched to imaging data, the mechanism of injury was clarified and divided into contact and noncontact injuries according to the direct impact on the lower limbs. Also, in terms of lower limb direction at the time of the event, contact injuries were categorized as hyperextension, varus, and valgus; noncontact injury was categorized as change in lower limb direction, e.g., turning the knee-pivoting, falling down with the knee in varus or valgus [21, 24–27]. The duration of activity was divided into 30-minute segments to account for the relationship between the duration of activity and the occurrence of injury. Concomitant injuries such as chondral lesions, meniscal injuries, and other ligamentous lesions were also assessed by MRI. All data were collected and recorded by an orthopedic resident and supervised by the attending physician.

## Statistical analysis

For the analysis, after data collection was finished, they were assessed from March 2023 to April 2023 for research purpose. According to the type 1 error value and accuracy equal to 5%, as well as considering the value of 50% for the percentage of knee ligament damage (in order to estimate the maximum sample size), at least 384 people should be examined in this study. In this research, a power calculation was conducted to determine the sample size that provided an estimated total number of 400 participants with knee related complaints. ACL-injured participants were primarily divided into two groups: Officers and Soldiers. Mechanisms of ACL injury, type of activity at the time of injury, and other factors related to injury were compared between the two groups of soldiers and officers. The statistical difference in demographic data between groups was analyzed with t tests. Frequencies were calculated; mechanisms of injury between the enlisted and officer groups were analyzed with the chi-square test. When the expected number of subjects in the cell of the frequency table is $\leq 5$, Fisher's exact test is used instead of the chi-square test. The one-way analysis of variance (ANOVA) was also used for multiple comparisons between the two groups. Data analysis was performed using SPSS 16.0 (SPSS Inc., Chicago, IL, USA) as the normality of the data set was tested by **Shapiro-Wilk test and** with a P value of less than 0.05 as the significance level.

## Results

A total of 402 patients were presented with knee injury. ACL tear accounted for 70.9% (285 out of 402 participants) of the diagnoses (Fig 1). Of the total 285 ACL-injured cases, 164 soldiers and 121 officers aged 18-51years (Mean±SD: 28.20±7.10) were considered for further analysis. The value for BMI was 21.22–32.87 kg/m$^2$ (Mean±SD: 25.26±1.76). Married personnel constituted 110 (38.6%) of total participants. There was a significant difference in age, BMI, marital status, and also injury pattern. The average age of soldiers was about eight years younger than that of officers (P = 0.001); among officers, BMI was statistically lower than that of soldiers (P = 0.001). Soldiers were more likely to be single, but the majority of officers were married; the proportion who were single was higher (P = 0.001). The highest proportion of injury patterns consisted of noncontact injuries (86.3%), or 246 of 285 injuries; for enlisted personnel, 148 of 164 (90.2%); for soldiers, 98 of 121 (80.9%). In both groups of officers and enlisted soldiers, the proportion of noncontact injuries was statistically higher (P = 0.001). Of the total 86.3%, the predominant lower limb status at the time of injury was knee rotation-pivoting (154/285; 54%); the next was falling with the knee in valgus position (59/285; 20.7%). Detailed data on participant demographics and mechanisms are provided in Table 1.

Our results showed that most injuries (182/285; 63.9%) occurred during military training exercises and 32.6%-93 out of 285- occurred during sports activities—e.g. any kind of sports such as soccer, volleyball, etc.—as well as during rest periods in the military environment. In addition, the incidence of injuries during military exercises was higher among soldiers than among officers (P = 0.002; Table 2).

The results also conveyed that there was a significant relationship between injury pattern and activities; among all activities, noncontact events accounted for a much larger proportion in both military activities and sports (P = 0.001). For injuries outside of the military environment—such as at home, during leisure time, etc.—however, most injuries occurred as contact sports and activities. The highest rate of injury occurred while twisting the knee during military activities (108/182: 59.3%), followed by sports activities (44/93; 47.3%) (Table 3).

Most injuries occurred in the first 30 minutes of an activity, in total, officer and soldier groups that represented 82.1%—234/285, 80.2%—97/121, and 83.5%—37/164, respectively (P = 0.463). In addition, the most common "time part of the day" when the injury occurred was before noon (63.9%-182 of 285 total injuries) (P = 0.287); however, because the P value is reported, these two relationships were not statistically significant (Table 4).

Regarding the dominance of the knee side, the right knee was more frequently injured in 182 of 285 (63.9%). The majority of the side of injury was also right, 175 of 285 (61.4%).

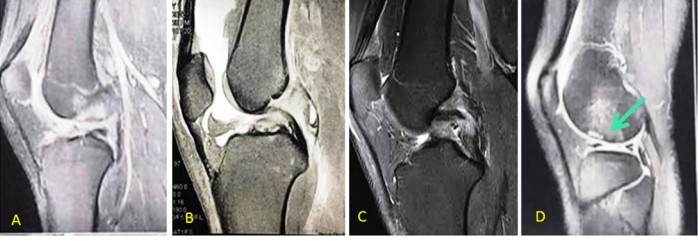

**Fig 1.** T-2 weighted MRI from some of participants depicting the knee ligament, meniscus and chondral injuries; A: Sagittal view of an anterior cruciate ligament (ACL) injury, B: Sagittal view of a medial meniscus bucket handle tear, C: Sagittal view of a posterior cruciate ligament (PCL) injury, D: Sagittal view of a chondral lesion at femoral condyle (green arrow). [*Extracted from ©Persian Gulf PACS viewer DICOM 0.3-in print*].

**Table 1. Participants' demographic data and mechanism of injury- Chi-squared test was done to calculate p value.**

| | Soldier (number = 164, 57.5%) | Officer (number = 121, 42.5%) | Sum of rows (number = 285, 100%) | *P* value |
|---|---|---|---|---|
| Age (Mean±SD) | 24.76±3.91 | 32.85±7.79 | 28.2±7.10 | 0.001 |
| BMI* (Mean±SD) | 25.59±1.91 | 24.81±1.43 | 25.26±1.76 | 0.001 |
| Marital status | | | | 0.001 |
| Single | 141(86) | 34(28.1) | 175(61.4) | |
| Married | 23(14) | 87(71.9) | 110(38.6) | |
| Dominant side | | | | 0.496 |
| Right | 102(62.2) | 80(66.1) | 182(63.9) | |
| Left | 62(37.8) | 41(33.9) | 103(36.1) | |
| Injured side | | | | 0.941 |
| Right | 101(61.6) | 74(61.2) | 175(61.4) | |
| Left | 63(38.4) | 47(38.8) | 110(38.6) | |
| Contact | 16(9.8) | 23(19) | 39(13.7) | 0.001 |
| Varus | 0(0.0) | 8(6.6) | 8(2.8) | |
| Valgus | 16(9.8) | 8(6.6) | 24(8.4) | |
| Hyper-extension | 0(0.0) | 7(5.8) | 7(2.5) | |
| Non-contact | 148(90.2) | 98(80.9) | 246 (86.3) | |
| Changing | 96(58.5) | 58(47.9) | 154(54) | |
| direction(pivoting) | 19(11.6) | 14(11.6) | 33(11.6) | |
| Fall in varus | 33(20.1) | 26(21.5) | 59(20.7) | |
| Fall in valgus | | | | |
| Sum of columns | 164(100) | 121(100) | 285(100) | |

* Body Mass Index

However, the difference between enlisted men and officers was not statistically significant for either dominance (P = 0.496) or injury side (P = 0.941) (Table 1).

Concomitant injuries were assessed by MRI and categorized as meniscal, cartilage, and other ligamentous injuries. Totally, the concomitant injuries were meniscal tears (135 out of 285; 47.4%), chondral lesions (98 out of 285; 34.5%), and damage to the posterior cruciate ligament and collateral ligaments (2 out of 285; 0.7% and 1 out of 285; 0.4%, respectively).

## Discussion

In this report, we document the high prevalence of ACL injuries among military personnel. We also found several noteworthy findings. The most common mechanism for ACL injury in the military was noncontact injury, most frequently by pivoting of the knee and then falling with the knee in valgus position. In addition, most ACL injuries occurred during sports

**Table 2. Association of the type of the activity while injury between soldiers and officers- Chi-squared test was done to calculate p value.**

| | Soldier (Number (%)) | Officer (Number (%)) | Sum of row (Number (%)) | P value |
|---|---|---|---|---|
| Military training (Combat, Marching, Walking Boarding, etc.) | 115(70.1) | 67(55.4) | 182(63.9) | 0.002 |
| Sport activity (Soccer, Volleyball, etc.) | 48(29.3) | 45(37.2) | 93(32.6) | |
| Outside activities* | 1(0.6) | 9(7.4) | 10(3.5) | |
| Sum of the column | 164(100) | 121(100) | 285(100) | |

* Outside refers to any activity that occurs outside military environment in non-military role, personal recreation or activity.

**Table 3. Association of the injury pattern and the type of activity—Chi-squared test was done to calculate p value.**

| | Military (Number (%)) | Sport (Number (%)) | Outside* (Number (%)) | Sum of rows (Number (%)) | P value |
|---|---|---|---|---|---|
| Contact | 7(3.8) | 24(26) | 8(80) | 39(13.7) | 0.001 |
| Varus | 0(0.0) | 7(7.5) | 1(10) | 8(2.8) | |
| Valgus | 7(3.8) | 17(18.3) | 0(0.0) | 24(8.4) | |
| Hyperextension | 0(0.0) | 0(0.0) | 7(70) | 7(2.5) | |
| Non-contact | 175(96.2) | 69(74) | 2(20) | 246(86.3) | |
| Change direction(pivoting) | 108(59.3) | 44(47.3) | 2(20) | 154(54) | |
| Fall in varus | 16(8.8) | 18(19.4) | 0(0.0) | 34(11.9) | |
| Fall in valgus | 51(28) | 7(7.5) | 0(0.0) | 58(20.4) | |
| Sum of columns | 182 (100) | 93 (100) | 10 (100) | 285 (100) | |

* Outside refers to any activity that occurs outside military environment in non-military role, personal recreation or activity.

activities when the knee was rotated. Although the most common time period to report an injury was within the first 30 minutes of activity and also during the before noon part of the day, this association was not statistically significant. Moreover, in terms of leg dominance and injured side of the lower extremity, the right knee was the most common injured side among soldiers and officers and overall, although with non-significant values.

Our results showed a prevalence rate of 70.9% for ACL injuries to the knee in military personnel, which is particularly close to a previous military study in Iran with a reported incidence of 77.8% for ACL injuries [37].

The current study found that ACL injuries occur more frequently by non-contact mechanisms and when the knee is pivoted. This is consistent with the findings of previous sports studies that most ACL injuries occur through non-contact injuries due to repetitive loading and fatigue and on landing [17, 24, 26, 38–41]. In addition, our study found that ACL injuries were more likely to be associated with military activities than sports, which is different from previous reports that found such injuries to be associated with military training rather than sports [11, 42]. However, another study reported that physical exercises and combat activities were the most common causes of injuries [43]. This may be due to the fact that the content of military and sports activities are somewhat the same in terms of the level of demand and effort-running, jumping, climbing, battling, struggling on the ball or subject; however, military exercises are thought to be mainly similar to the training of professional athletes in order to maintain the expected physical level [1]. According to previous reports from the sports field, prevention plans have reduced the incidence of ACL injuries to approximately 50–80% [15, 17, 34–36]. Therefore, it could be suggested that the application of similar injury prevention plans would reduce the incidence of ACL injuries in the military setting as well [17, 44].

**Table 4. Time period of the injury occurrence between soldiers and officers- Chi-squared test was done to calculate p value.**

| | Soldier (Number (%)) | Officer (Number (%)) | Sum of rows (Number (%)) | P value |
|---|---|---|---|---|
| 0–30 minutes | 137(83.5) | 97(80.2) | 234(82.1) | 0.463 |
| After 30 minutes | 27(16.5) | 24(19.8) | 51(17.9) | |
| Sum of columns | 164(100) | 121(100) | 285(100) | |
| Before-noon | 109(66.5) | 73(60.3) | 182(63.9) | 0.287 |
| Afternoon | 55(33.5) | 48(39.7) | 103(36.1) | |
| Sum of columns | 164(100) | 121(100) | 285(100) | |

We observed that the maximum incidence of injury, although not statistically significant, was within the first 30 minutes of activity onset and also in the periods before noon rather than in the afternoon. This differs from the results of a previous study that showed a higher prevalence of injury in the first 30–60 minutes after activity onset [11]. Our differing result could be due to inadequate muscle preparation for activities. Neuromuscular warm-up has been shown to prepare muscles and reduce the risk of ACL injury by improving muscular firing patterns and thus knee stabilizing muscles, potentially reducing the risk of ACL injury. Since soldiers participate intensively in training, including competitive sports and combat performance, there would be tremendous fatigue and repetitive strain in addition to unprepared and unwarmed muscles at the beginning of activities [41, 45–47]. In our study, there was also a significant correlation for age, marital status, and BMI between groups of soldiers and offices in multiple comparisons, which is consistent with the findings of previous reports [48–51]. Regarding lower limb dominant and injured sides available published report were limited in the literature; a previous report showed that of the 62 patients 36 (58%) had a left-side injury and 26 (42%) had a right-side ACL injury. Leg dominance, defined by the leg preferred for kicking a ball, was recorded for 46 patients where 43 had a right-sided dominance [52]; their results in terms of proportion were similar to our finding with a higher proportion for right sides, although our results were not statically significant. However, another study in Iranian athletes reported no differences in knee injuries between the limb non-dominant and dominant sides [29].

It is crucial that decision makers recognize the factors associated with ACL injuries in order to develop practical strategies to reduce injuries [23, 29]. Generally, it may be important to consider demographic characteristics of individuals, and type of activities and mechanisms, and careful positioning of the lower leg in addition to the timing and duration of activities when caring for ACL injuries. Preventive measures that reduce the introduced related factors to ACL injuries may hypothetically reduce the incidence of ACL injuries in the military setting. The findings of this study may serve as a preliminary study for future research to develop strategies including training, and dress and equipment modifications, to prevent ACL injuries in the military setting and to adapt the military environment to reduce the incidence of military knee injuries.

Our study had several limitations. First, it was a cross-sectional analytic study based on medical records and imaging and self-reported data. In addition, there was no standardized control group for comparison. Second, we studied all ACL/meniscus/cartilage injuries, including partial and complete injuries and functional and non-functional injuries, all of which have an important impact on outcomes and injury burden and decision making when preferably studied in detail and separately, especially the ACL injuries. In addition, we did not analyzed subgroups of military trainings and sport activities separately. Third, there were some environmental factors in the military environment that were difficult to control, including dressings and equipment worn, the ground surface, etc., and it was not possible to fully reproduce all variables in the final assessment. Thus, further cohort studies with control groups for a detailed comparison of all types of soft tissue injuries in the knee, as well as a breakdown by sports and military training activities, and controlling for confounders and other environmental and equipment-related factors for injury occurrence would provide more evidence.

## Conclusion

The findings support those of previous studies, that in military personnel, the ACL injury is one the most common knee injuries. Also, the main mechanism for ACL injuries in the military is noncontact events, most frequently while pivoting of the knee. The most affected

personnel were soldiers; most injuries occurred during military-related activities rather than playing a sport in military environment. The results need to be confirmed in future larger scale multicenter studies.

The findings support those of previous studies, that in military personnel, the most common knee injury is damage to the ACL. The most common mechanism for ACL tears was noncontact events, specifically twisting, during military activities rather than sports. These findings on mechanisms and prevalence of injury may help develop ACL injury prevention programs that may minimize the occurrence of such injuries in the military setting.

## Supporting information

**S1 Data.**
(ZIP)

## Acknowledgments

The authors are appreciated to all participants for their cooperation. Also, the authors would like to thank the Clinical Research Development Unit of Baqiyatallah Hospital, for all their support and guidance through conducting this study. Finally, the authors wish to express special thanks to Dr. Mohsen SaberiIsfeedvajani M.D., for his consulting on data analysis.

## Author Contributions

**Conceptualization:** Seyyed-Mohsen Hosseininejad, Mohammad Kazem Emami Meybodi.

**Data curation:** Seyyed-Mohsen Hosseininejad.

**Formal analysis:** Seyyed-Mohsen Hosseininejad, Mehdi Raei.

**Methodology:** Seyyed-Mohsen Hosseininejad, Mohammad Kazem Emami Meybodi, Mehdi Raei.

**Software:** Mehdi Raei.

**Supervision:** Alireza Rahimnia.

**Writing – original draft:** Seyyed-Mohsen Hosseininejad, Mohammad Kazem Emami Meybodi, Mehdi Raei, Alireza Rahimnia.

**Writing – review & editing:** Seyyed-Mohsen Hosseininejad, Mohammad Kazem Emami Meybodi, Mehdi Raei, Alireza Rahimnia.

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
