## [Decision Letter · Decision Letter 0]

28 Jul 2023

PONE-D-23-13460Anterior Cruciate Ligament Tear in the Iranian Military Setting; the Injury Prevalence and MechanismPLOS ONE

Dear Dr. Rahimnia,

Thank you for submitting your manuscript to PLOS ONE. After careful consideration, we feel that it has merit but does not fully meet PLOS ONE’s publication criteria as it currently stands. Therefore, we invite you to submit a revised version of the manuscript that addresses the points raised during the review process.

We look forward to receiving your revised manuscript.

Kind regards,

Amit Joshi, MD

Academic Editor

PLOS ONE

Additional Editor Comments:

dear author.

our reviewer have suggested some correction on your manuscript. kindly address ech point.

Reviewers' comments:

Reviewer's Responses to Questions

**Comments to the Author**

1. Is the manuscript technically sound, and do the data support the conclusions?

Reviewer #1: Partly

Reviewer #2: Partly

2. Has the statistical analysis been performed appropriately and rigorously? 

Reviewer #1: No

Reviewer #2: No

3. Have the authors made all data underlying the findings in their manuscript fully available?

Reviewer #1: Yes

Reviewer #2: Yes

4. Is the manuscript presented in an intelligible fashion and written in standard English?

Reviewer #1: Yes

Reviewer #2: Yes

5. Review Comments to the Author

Reviewer #1: Reviewer’s Comments

Anterior Cruciate Ligament Tear in the Iranian Military Setting; the Injury Prevalence and Mechanism

Comments for the Author

Line 22 The introduction lacks a scientific rationale for this study.

Line 23 The author should provide a specific hypothesis.

Line 33 Could all the patients in this study recall the mechanism of injury? E.g., Fall with the knee in valgus position.

Line 35 Difference between military exercises and athletic exercise?

Line 42 Could you please elaborate the prevention strategies this study has helped to develop?

Line 82 Please mention the rationale for excluding the chronic injury.

Line 85 Clinical and imaging findings for ACL tear have inter- observer and intra- observer variability. If only those patients taken for arthroscopic surgery were studied and intraoperative findings considered the gold standard, the results would be more reproducible.

Line 98 Clinical and imaging findings with respect to chondral and meniscus injuries have inter- observer variability.

Line 128 Please define the military exercise.

Line 176 Please clarify the similarities and differences between the military and sports activities.

Line 216 This study does not determine the exact measures to prevent an injury to ACL.

Line 192 The author has assumed that none of the patients were exposed to warm-up exercises.

Line 194 A gender sub-analysis could give a better insight.

Line 194 Besides the mechanism of injury, there are other anatomical factors responsible for ACL injury. Please clarify how these factors have been matched.

Earlier studies have concluded even more detailed mechanisms of injury and prevention strategies. Please elaborate what this study adds extra to the existing knowledge related to the topic.

Reviewer #2: Authors should be congratulated for such a nice work. It definitely has scientific merit and will be an interesting topic for readers. However, here are some suggestions:

introduction section: please incorporate paragraphs 1 and 2 to shorten the introduction section and make the background information more relevant. line 57-59 can be deleted because it is not contributing much to the flow of writing.

methodology section: 1. Please mention the sample size calculation and sampling technique in detail (provide formula).

2. Please mention whether or not the normality of the data set was tested?

3. Please justify the use of statistical test based on the normality of the data. Consult statistician.

4. Please revise the statement mentioned in line 85/86, as it is confusing that you enrolled 285 or 402 patients. Better to mention it in the result section with flow of participants.

5. As it is a cross-sectional descriptive analytical study, providing confidence intervals would be better.

6. You have mentioned that you included patients with knee complaints with pain, giving away, and locking. Did that mean all patients had locking symptoms? please clarify.

Result section: 1. please avoid explanation of the study findings. for e.g. significantly lower. Please provide study findings as it is and explain them in the discussion section. All the tables are self-explanatory, so, limited writing is sufficient. In addition, please provide which test was done to calculate p value in all 4 table legends.

Discussion section: it is weak and can be improved significantly by careful writing and direct comparison with previous study findings.

1. Please follow simple rule of academic writing that 1 paragraph should evaluate 1 outcome. Use format of breaking sentence, data, evaluation, analysis with relevant previous study findings, reflection (if applicable), and closing sentence.

2. Please do not create new information which was not mentioned in the objective and methodology. for eg., correlation (in line 194), I did not find anywhere it was mentioned.

3. please delete line 156-164 as these statements are general remarks. Can be used in introduction section though.

4. please compare and contrast your findings with specific studies rather than in general with multiple citations.

conclusion: please delete line 213-215, as it is not in the scope of the study. instead, it would be better to write the implications of the findings. for eg. such prevalence of ACL injury would.....

6. PLOS authors have the option to publish the peer review history of their article (what does this mean?). If published, this will include your full peer review and any attached files.

Reviewer #1: No

Reviewer #2: **Yes: **Subhash Regmi

---

## [Author Response · Author response to Decision Letter 0]

23 Feb 2024

Dear Editor, 

Thank you for evaluating my work. The following comments have been check and edited in the manuscript file. 

Line 22 the introduction lacks a scientific rationale for this study. Edited 

Line 23 The author should provide a specific hypothesis. Edited 

Line 33 Could all the patients in this study recall the mechanism of injury? E.g., Fall with the knee in valgus position. Yes, as our sample included acute and new cases with knee injury they could recall.

Line 35 Difference between military exercises and athletic exercise? Added and Defined in the methods

Line 42 Could you please elaborate the prevention strategies this study has helped to develop? As mentioned in discussion and conclusion: The findings on mechanisms and prevalence of injury may help develop ACL injury prevention programs that may minimize the occurrence of such injuries in the military setting.

Line 82 Please mention the rationale for excluding the chronic injury. The most important rationale was the recall bias which may occur with a longer time since injury. Other reasons include narrowing diagnosis and associated injuries.

Line 85 Clinical and imaging findings for ACL tear have inter- observer and intra- observer variability. If only those patients taken for arthroscopic surgery were studied and intraoperative findings considered the gold standard, the results would be more reproducible. Yes, right, we did all investigation by one person to reduce any bias and error but included all injured knees in the study.

Line 98 Clinical and imaging findings with respect to chondral and meniscus injuries have inter- observer variability. This also had been investigated by the same person.

Line 128 Please define the military exercise. Defined previously 

Line 176 Please clarify the similarities and differences between the military and sports activities. As mentioned, both activities-as mentioned the examples- are the same in terms of magnitude of activity the difference are in kind of activity like combat and sport activities ,etc.

Line 216 This study does not determine the exact measures to prevent an injury to ACL. as we discussed, our study provided data on knowing mechanism and prevalence of the injury which could be taken into account for neutralizing the associated factors of injury or decrease chance of getting injured.

Line 192 The author has assumed that none of the patients were exposed to warm-up exercises.no we mean that although participants will get involve in warm-up exercise before actual activity, at the beginning of the actual and main activity, there would be a greater chance of injury as the body system has just faced real and serious activity, physiologically.

Line 194 A gender sub-analysis could give a better insight. Yes, it was better if more extensive literature were available. we tried to elaborate it anyway.

Line 194 Besides the mechanism of injury, there are other anatomical factors responsible for ACL injury. Please clarify how these factors have been matched. Actually our study was a cross sectional analytic study and we did not do any classic matching.

Earlier studies have concluded even more detailed mechanisms of injury and prevention strategies. Please elaborate what this study adds extra to the existing knowledge related to the topic. We conducted the study in a referral military Iranian hospital which no previous data have been reported.so geographically; our data will help know regional data.

Reviewer #2: Authors should be congratulated for such a nice work. It definitely has scientific merit and will be an interesting topic for readers. However, here are some suggestions:

introduction section: please incorporate paragraphs 1 and 2 to shorten the introduction section and make the background information more relevant. line 57-59 can be deleted because it is not contributing much to the flow of writing. Edited 

methodology section: 1. Please mention the sample size calculation and sampling technique in detail (provide formula). According to the type 1 error value and accuracy equal to 5%, as well as considering the value of 50% for the percentage of knee ligament damage (in order to estimate the maximum sample size), at least 384 people should be examined in this study. In this research, the researcher studies about 402 patient by Census sampling in which 285 patients had ACL injury and included for analysis.

2. Please mention whether or not the normality of the data set was tested? Yes did in SPSS.

3. Please justify the use of statistical test based on the normality of the data. Consult statistician. Checked.

4. Please revise the statement mentioned in line 85/86, as it is confusing that you enrolled 285 or 402 patients. Better to mention it in the result section with flow of participants. Edited 

5. As it is a cross-sectional descriptive analytical study, providing confidence intervals would be better.

6. You have mentioned that you included patients with knee complaints with pain, giving away, and locking. Did that mean all patients had locking symptoms? please clarify. No, any of the symptoms could be a presenting symptoms.

Result section: 1. please avoid explanation of the study findings. for e.g. significantly lower. Please provide study findings as it is and explain them in the discussion section. All the tables are self-explanatory, so, limited writing is sufficient. In addition, please provide which test was done to calculate p value in all 4 table legends. - Chi-squared test was done to calculate p value

Discussion section: it is weak and can be improved significantly by careful writing and direct comparison with previous study findings. Edited :

1. Please follow simple rule of academic writing that 1 paragraph should evaluate 1 outcome. Use format of breaking sentence, data, evaluation, analysis with relevant previous study findings, reflection (if applicable), and closing sentence.

2. Please do not create new information which was not mentioned in the objective and methodology. for eg., correlation (in line 194), I did not find anywhere it was mentioned. It is mentioned in Table 1 and paragraph 1 in result. 

3. please delete line 156-164 as these statements are general remarks. Can be used in introduction section though. 

4. please compare and contrast your findings with specific studies rather than in general with multiple citations. 

conclusion: please delete line 213-215, as it is not in the scope of the study. Instead, it would be better to write the implications of the findings. for eg. such prevalence of ACL injury would..... .edited. 

Alireza Rahimnia

Regards, 

September 27, 2023

---

## [Decision Letter · Decision Letter 1]

12 Apr 2024

PONE-D-23-13460R1Prevalence and Mechanisms of Anterior Cruciate Ligament Tears in Military Personnel: A Cross-Sectional Study in IranPLOS ONE

Dear Dr. Rahimnia,

Thank you for submitting your manuscript to PLOS ONE. After careful consideration, we feel that it has merit but does not fully meet PLOS ONE’s publication criteria as it currently stands. Therefore, we invite you to submit a revised version of the manuscript that addresses the points raised during the review process. Some minor comments which need to be addressed are reported below.

We look forward to receiving your revised manuscript.

Kind regards,

Luciana Labanca

Academic Editor

PLOS ONE

Journal Requirements:

Additional Editor Comments:

Lines 21-23. The meaning of the first sentence of the abstract is not clear. Please rephrase it.

Line 90. Please, move the approval number after the name of the Institution providing the approval.

Line 96. There are two dashes which should be removed.

Line 121. Please change ; with .

Line 125. Please change ; with .

Line 149. Remove one of the dashes

Line 254. In addition

Line 263. Since you did not assess and reported other kind of injuries, I suggest to change in “…the ACL injury is one the most common knee injuries.”

Reviewers' comments:

Reviewer's Responses to Questions

**Comments to the Author**

1. If the authors have adequately addressed your comments raised in a previous round of review and you feel that this manuscript is now acceptable for publication, you may indicate that here to bypass the “Comments to the Author” section, enter your conflict of interest statement in the “Confidential to Editor” section, and submit your "Accept" recommendation.

Reviewer #2: All comments have been addressed

2. Is the manuscript technically sound, and do the data support the conclusions?

Reviewer #2: Yes

3. Has the statistical analysis been performed appropriately and rigorously? 

Reviewer #2: Yes

4. Have the authors made all data underlying the findings in their manuscript fully available?

Reviewer #2: Yes

5. Is the manuscript presented in an intelligible fashion and written in standard English?

Reviewer #2: Yes

6. Review Comments to the Author

Reviewer #2: (No Response)

7. PLOS authors have the option to publish the peer review history of their article (what does this mean?). If published, this will include your full peer review and any attached files.

Reviewer #2: **Yes: **Subhash Regmi

---

## [Author Response · Author response to Decision Letter 1]

22 Apr 2024

Dear Editor-in-chief

Thank you for evaluating my work. All the helpful comments were reviewed and edited in the text as follows:

The reference list is correct and in sync with the text.

Lines 21-23. The meaning of the first sentence of the abstract is not clear. Please rephrase it. It is checked and revised.

Line 90. Please, move the approval number after the name of the Institution providing the approval. Checked , edited.

Line 96. There are two dashes which should be removed. Edited.

Line 121. Please change ; with . changed.

Line 125. Please change ; with . changed

Line 149. Remove one of the dashes. Removed.

Line 254. In addition. Edited.

Line 263. Since you did not assess and reported other kind of injuries, I suggest to change in “…the ACL injury is one the most common knee injuries.” Edited.

Regards,

Monday, April 22, 2024

---

## [Editor Report · Decision Letter 2]

24 Apr 2024

Prevalence and Mechanisms of Anterior Cruciate Ligament Tears in Military Personnel: A Cross-Sectional Study in Iran

PONE-D-23-13460R2

Dear Dr. Rahimnia,

We’re pleased to inform you that your manuscript has been judged scientifically suitable for publication and will be formally accepted for publication once it meets all outstanding technical requirements.

Kind regards,

Luciana Labanca

Academic Editor

PLOS ONE
---

## [Editor Report · Acceptance letter]

14 May 2024

PONE-D-23-13460R2 

PLOS ONE

Dear Dr. Rahimnia, 

I'm pleased to inform you that your manuscript has been deemed suitable for publication in PLOS ONE. Congratulations! Your manuscript is now being handed over to our production team.

Kind regards, 

on behalf of

Dr. Luciana Labanca 

Academic Editor

PLOS ONE